# Passive Continuous Variable Quantum Key Distribution through the Oceanic Turbulence

**DOI:** 10.3390/e25020307

**Published:** 2023-02-07

**Authors:** Yiwu Zhu, Lei Mao, Hui Hu, Yijun Wang

**Affiliations:** School of Automation, Central South University, Changsha 410083, China

**Keywords:** quantum key distribution, seawater channel, quantum communications

## Abstract

Continuous variable quantum key distribution (CVQKD) can be potentially implemented through seawater channels, whereas the involved oceanic turbulence has a negative effect on the maximal transmission distance of quantum communication systems. Here, we demonstrate the effects of the oceanic turbulence on the performance of the CVQKD system and suggest an implementation feasibility of the passive CVQKD through the oceanic turbulence-based channel. We achieve the channel transmittance characterized by the transmission distance and depth of the seawater. Moreover, a non-Gaussian approach is used for performance improvement while counteracting the effects of excess noises on the oceanic channel. Numerical simulations show that the photon operation (PO) unit can bring reductions of excess noise when taking into account the oceanic turbulence, and hence results in performance improvement in terms of transmission distance and depth as well. The passive CVQKD explores the intrinsic field fluctuations of a thermal source without using an active scheme and hence has a promising application in chip integration for portable quantum communications.

## 1. Introduction

Continuous variable quantum key distribution (CVQKD) is a branch of quantum communications [1,2], where two participants called Alice and Bob encode information symmetrically with quadratures ($\hat{x}$ and $\hat{p}$) of the optical field with Gaussian modulation and then establish a string of secret key through public channels. It can make use of continuous light for the source preparation and the homodyne detector or heterodyne detector for detection [3,4]. Consequently, it has an advantage that can be potentially implemented with the existing optical technology, which gives birth to perfect compatibility and usability in modern optical communications. In implementations of the traditional CVQKD, Gaussian-distributed random numbers *x* and *p* are generated and coherent states |x+ip〉 is prepared with amplitude and phase modulations. The security of practical CVQKD systems has been proved in theoretical and practical frameworks [5]. Moreover, it can be used for potential implementations in quantum computers [6] and quantum networks [7]. The adaptability of CVQKD is one of its great attractions while holding up the malicious invasion in modern communication systems [8]. However, the quantum random number generator (QRNG) is required so that high-speed CVQKD at the chip level may have some disadvantages in terms of performance and size parameter trade-offs in most integrated photonics circuits.

Currently, kinds of CVQKD protocols have been suggested in free space, such as satellite-to-satellite links, satellite-to-ground links, air-to-water channels, and so on [9]. Unfortunately, the transmission coefficient fluctuates due to the effects of turbulence in free space, where the coherent detection may be distorted, leading to the decreased performance of the quantum communication system [10,11]. To model the turbulence in free space, several scenarios of turbulence have already been proposed [12,13], including beam wandering and elliptic-beam model, which demonstrates the evolution of the receiving beam in weak and strong atmospheric turbulence. However, there is no suitable channel model that can illustrate the effects of the turbulence on performance of the free-space CVQKD system.

To overcome the aforementioned disadvantages, a passive state preparation can be used for CVQKD with the thermal source, where the QRNG has been replaced by a thermal source, beam splitters, optical attenuators and homodyne detectors [14]. It offers a prospect of simple integration with low cost which is much more suitable for portable applications [15,16]. We will illustrate characteristics of the turbulence-modulated channel that can be used to describe the effect of oceanic turbulence on the performance of the passive CVQKD. We characterize the structure of the passive CVQKD through the seawater channel and demonstrate the effect of the oceanic turbulence on the performance of the system, which involves the intrinsic field fluctuations of a thermal source. Different from the traditional CVQKD, it does not require active modulations and hence has a potential application in chip integration and portable quantum communications. Moreover, we counteract the effect of the channel-added noise by employing a non-Gaussian photon operation (PO) unit at the receiver, which can efficiently compensate for the channel-added noise generated from the imperfect detector, leading to the performance improvement of the CVQKD system in the terahertz band.

This paper is organized as follows. In Section 2, we propose the scheme of PO-based passive CVQKD in terahertz band through oceanic turbulence in seawater channel. Given the oceanic turbulence, we illustrate the characteristics of the PO-embedded detector and demonstrate the effects of the PO unit on the CVQKD system in Section 3. In Section 4, we consider the performance analysis of the passive CVQKD system with numerical simulations. Finally, conclusions are drawn in Section 5.

## 2. Passive CVQKD through Seawater Channel

As mentioned in the traditional CVQKD [17,18,19], quantum states can be prepared actively, and hence usually called active CVQKD. It requires a highly precise modulator to meet the constraints of a complex modulation format while suppressing the modulation errors, making it difficult to lay a portable terminal device for practical implementations. In order to solve the above problems, we suggest an approach to the passive CVQKD through oceanic turbulence, which involves thermal states in the terahertz (THz) band, as shown in Figure 1.

First of all, Alice splits the output of the thermal source (TS) with a beam splitter (BS), resulting in two spatial modes $A_{1}^{\prime }$ and $A_{2}^{\prime }$. She chooses one mode $A_{2}^{\prime }$ and measures with results $\left(x_{2},p_{2}\right)$. Alice can get to know quadratures of $A_{2}^{\prime }$ by scaling down the measurement results by using an optical attenuator. After that, she performs the suitable optical attenuation on another mode $A_{1}^{\prime }$, resulting in mode $A_{1}$, which is transmitted to Bob through the seawater channel. Bob’s actions on mode $B_{1}$ can be performed by using the PO unit, with input mode $B_{1}$ and output mode $B_{2}$, as shown in Figure 1. There is a correlation between the measurement results of Alice and Bob. Finally, they can generate a secret key after performing the data post-processing.

Different from the active CVQKD, which reduces the modulation deviation with the high extinction ratio modulators, the passive CVQKD makes full use of a thermal state for state preparation, rather than performing Gaussian modulation to a coherent state, and hence it simplifies implementations of the CVQKD system in practice. In Gaussian modulated coherent states (GMCS) CVQKD, the randomness comes from a true random number generator, whereas in the passive CVQKD, the randomness comes from the correlated thermal states that are split from a common thermal source. The former allows for the entanglement source that can be controlled by Eve while the latter requires a trusted source. In addition, Alice makes use of a high extinction ratio modulator to adjust the phase and amplitude of quantum states, which makes it difficult to realize in practice.

Moreover, due to the effects of the environmental thermal noise, the security threshold of the passive CVQKD is still high in terms of the secret key rate and the maximal transmission distance [20,21,22]. Fortunately, the turbulence under seawater can be efficiently compensated by deploying a suitable PO unit, such as photon catalysis, photon addition, and photon subtraction, to mitigate the turbulence wavefront aberrations of the received laser carrier signals. As shown in Figure 2, we find that zero photon catalysis (ZPC) performs better than others in terms of successful probability, and thus we select the ZPC-based OP unit for the performance improvement of the passive CVQKD system.

## 3. The PO Unit for Loss Compensation

To date, the passive CVQKD can be carried out with the PO-enabled unit through a seawater channel in terahertz (THz), and it turns out to be the potential to meet the needs of high-rate communications. In what follows, we will consider the effects of the PO unit on the passive CVQKD while taking into account the thermal source in the THz band.

The schematic diagram of the PO-embedded receiver for CVQKD can be shown in Figure 1. Alice prepares for the thermal states in the THz band. She performs the Gaussian modulation with two quadrature components, involving amplitude quadrature (*x*) and phase quadrature (*p*). The overall variance that includes the shot noise and modulation variance is given by $V=V_{A}+V_{0}$, where $V_{A}$ is the zero-centered Gaussian distributed modulation variance and $V_{0}$ is the shot noise described as $V_{0}=2{\bar{n}}_{0}+1$. The parameter ${\bar{n}}_{0}$ is represented as
(1)$${\bar{n}}_{0}={\left[\exp\left(h/\lambda \tau \right)-1\right]}^{-1},$$
where λ denotes the wavelength of the THz thermal source, *h* is the temperature, and τ is Boltzmann’s constant. The shot noise fluctuates due to thermal fluctuations and modulation. Alice measures the coherent states projected on one mode, while she projects another mode, transmitting through the untrusted seawater channel.

At the receiver, Bob measures weak quantum signal with the local oscillator (LO) pulse in a shot-noise-limited homodyne detector (HD) or herterodyne detector (HE). Upon receiving the transmitted pulses, he demultiplexes signal and LO pulses, where an amplitude modulator is applied on the signal path to randomly attenuate the amplitude for the shot-noise estimation. Before performing detection, a suitable PO unit, such as zero photon catalysis (ZPC), is applied on the signal path and LO path to measure and correct wavefront distortions to restore signal quality by flattening the distorted wavefront. After that, Bob performs optical coherent detection to estimate the excess noise.

In Figure 3, we show that the PO unit can be regarded as a filter for attenuation in practice. The quantum signals follow Gaussian distribution mixed with excess noise for the seawater channel. Followed by the PO unit, we find that the weakened signals can be efficiently recovered while using the homodyne detector at the receiver. We note that in a PO unit, a wavefront sensor is applied to correct distorted wavefronts by physically distorting the reflective face sheet using mechanical actuators. The PO combining electronics with optics can realize the detection of distorted wavefront and then correct distorted wavefront in real-time. In addition, the PO unit will not start working until a very short period of time due to the characteristics of the transmission of the thermal source through oceanic turbulence in the seawater channel.

## 4. Performance Analysis

### 4.1. Parameter Evaluation of Seawater Channel

Before demonstrating the performance of the PO-enabled CVQKD through oceanic turbulence, we perform numerical simulations to show the characteristics of the seawater channel, which may have an effect on the performance of the CVQKD system.

A seawater channel can be modulated with the probability of transmittance distribution, which can be described with Beer’s law [20,23]. When we ignore the multi-scattering of photons for the point-to-point propagation of signals, the deterministic losses caused by the ocean extinction have an effect on the transmittance with an estimation form [24]
(2)$$T_{B}=\exp\left[-\left(a(\lambda ,d)+b(\lambda ,d)\right)L\right],$$
where a(λ,d) and b(λ,d) denote the absorption coefficient and the scattering coefficient of light with wavelength λ, depth *D* and transmission distance *L*, which are shown in Figure 4a,b, respectively. Consequently, the transmittance can be described with Beer’s law, as shown in Figure 4c. The transmittance decreases exponentially with the increase of transmission distance, and depth as well. In addition, we have the characteristics that illustrate the transmittance as a function of depth and transmission distance, which can be described with numerical simulation using the Monte-Carlo method in Figure 4d. It is interesting to find that, for the given transmission distance, there is a depth interval (from 50 m to 70 m) with better performance of the transmittance, due to the fact that the total attenuation is small for this interval. Therefore, we obtain the optimal depth that can be used for numerical simulations to illustrate the performance improvement of the passive CVQKD system.

### 4.2. Performance Analysis of the PO-Enabled CVQKD System

In what follows, we illustrate the performance of the PO-enabled CVQKD system with passive state preparation. We achieve performance improvement while comparing with the traditional scheme in terms of transmission distance and depth as well.

As shown in Figure 5, we demonstrate the secret key rate of the PO-enabled CVQKD system as a function of transmission distance for the oceanic turbulence model at the depths 80 m, 120 m and 200 m. We find that there is a decline in the secret key rate with the increasing transmission distance. Compared with the performance of the traditional scheme, the PO-embedded unit at the receiver, which serves as an attenuation in essence, enables the performance improvement of the CVQKD system.

As shown in Figure 6, we demonstrate the secret key rate of the PO-enabled CVQKD system as a function of the depth for the oceanic turbulence model at the transmission distances 1.65 m, 4.5 m and 6.5 m. Compared with the traditional scheme, the PO-embedded scheme performs better when considering the performance improvement of the CVQKD system. There is also a decline and then an increase in the secret key rate at the depth of 120 m, which is a particularly new phenomenon for both the PO-enabled CVQKD and the traditional CVQKD system. For the above-mentioned two cases, we find that the secret key rate is higher at depth 1.65 m than that at depths 4.5 m or 6.5 m. The reason is that the ocean absorption $t_{abs}$ and the scattering factor $t_{sca}$ increase rapidly at depths of 100 m to 140 m, which is shown in Figure 6b. We note that in actual situations, phytoplankton, particulate minerals and other extinction factors are usually enriched at this depth interval. In addition, we note the channel-added noise is relative to the transmission distance and the depth. For the given depth, the long transmission distance results in large channel-added noise. The reason is that when the turbulence gives rise to the fluctuation of the refractive index, the wavefront of the arriving beam is undermined, which degrades the performance of the transmission beam by contributing to beam wandering and redistribution of energy across the beam section (beam broadening and deformation).

## 5. Conclusions

We have suggested an approach to performance improvement of the passive CVQKD through the oceanic turbulence in seawater channels, which can be improved by the embedded PO unit at the receiver. In addition, the characteristics of the seawater channel are illustrated with numerical simulations. We demonstrate the influences of the parameters of the seawater channel on the performance of the CVQKD system that involves the secret key rate, the transmission distance and the depth as well. We find that the embedded PO unit may have a positive effect on the performance of the passive CVQKD in free space. We illustrate the performance of the PO-enabled CVQKD through the seawater channel, which results in the increased secret key rate. It suggests an elegant approach to performance improvement of the practical CVQKD system in oceanic environments.

## Figures and Tables

**Figure 1 entropy-25-00307-f001:**
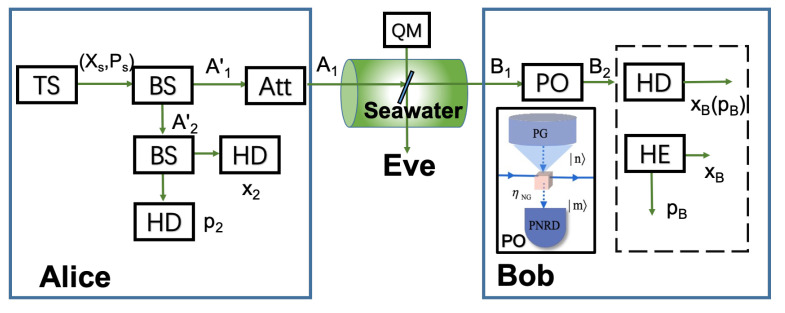
The schematic diagram of CVQKD with the passive state preparation. TS denotes thermal source, BS denotes beam splitter, Att is the optical attenuator, PO is photon operation involving photon generator (PG) with *n* input photons and *m* output photons, PNRD denotes photon number resolving detector, HD is the homodyne detector and HE is the heterodyne detector.

**Figure 2 entropy-25-00307-f002:**
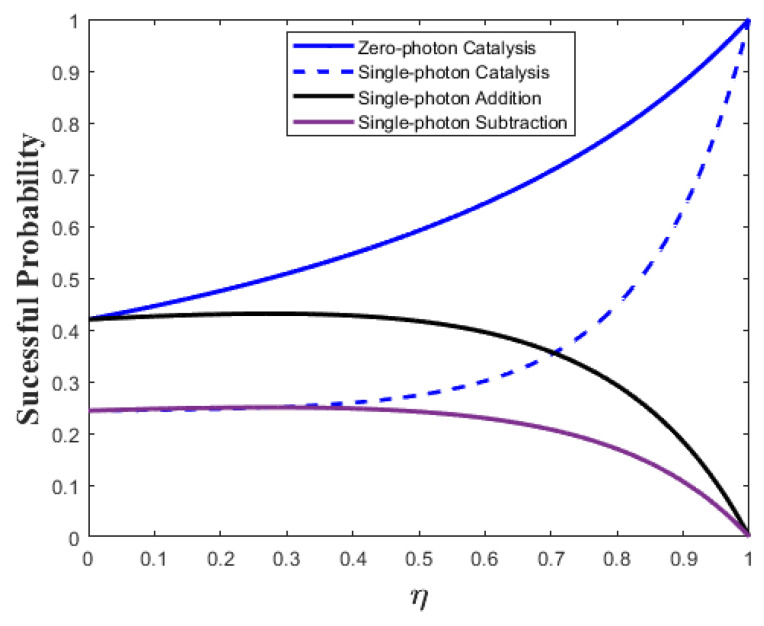
Successful probabilities of photon operations that involve photon catalysis, photon addition, and photon subtraction.

**Figure 3 entropy-25-00307-f003:**
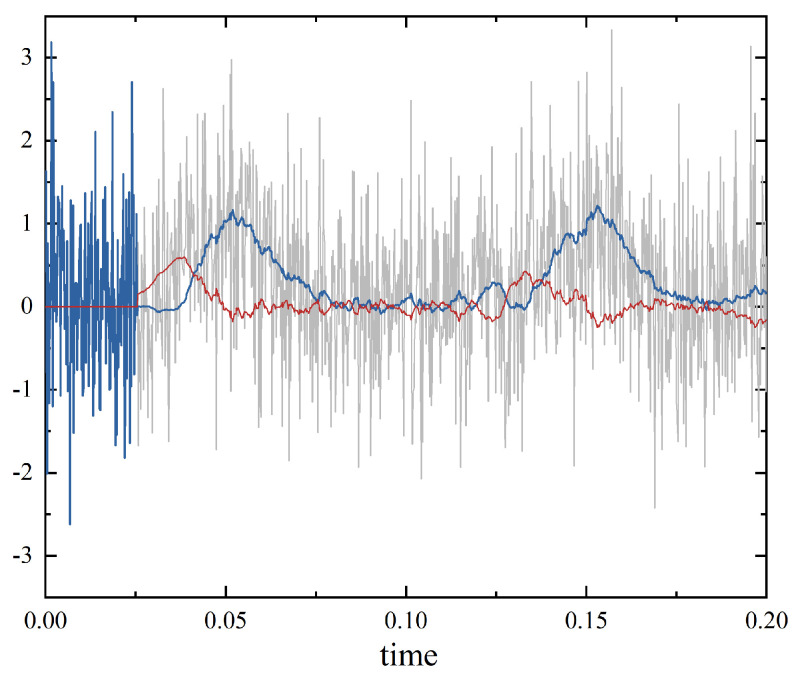
The effect of the ZPC-involved PO unit on the transmitted quantum signals through oceanic turbulence in the seawater channel. The gray line represents the waveform of the signal mixed with Gaussian-distributed noise. The blue line denotes the processed waveform of the signal mixed with Gaussian-distributed noise after using the PO unit. The red line denotes the signal error of the processed thermal source.

**Figure 4 entropy-25-00307-f004:**
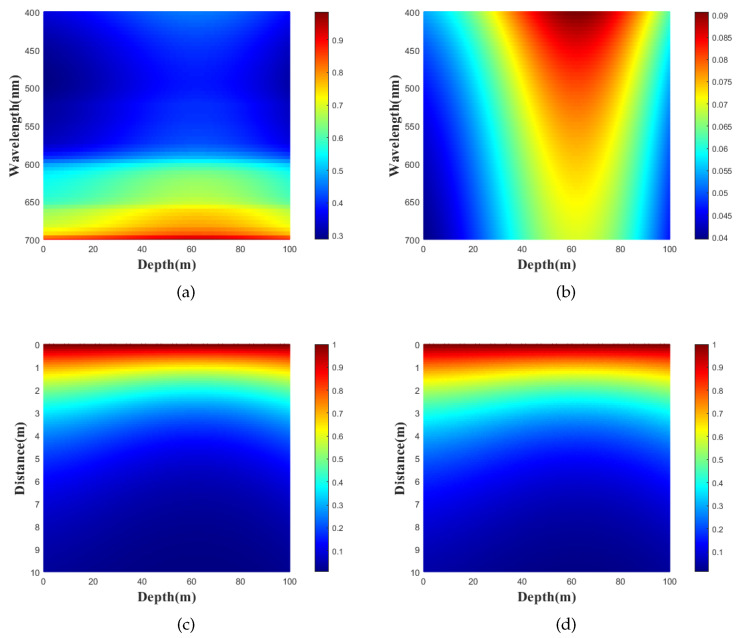
Parameters for transmittance in the seawater channel. (**a**) Variation of absorption coefficient a(λ,d), (**b**) Variation of scattering coefficient b(λ,d), (**c**) Transmittance $T_{B}$ achieved with Beer method, (**d**) Transmittance $T_{B}$ achieved with Monte-Carlo method.

**Figure 5 entropy-25-00307-f005:**
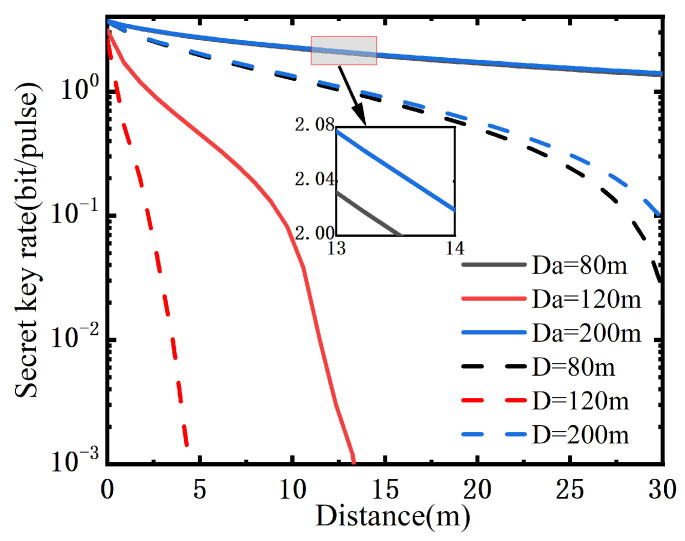
The secret key rate of the PO-enabled CVQKD system as a function of the transmission distance. The notation Da denotes the case of the PO-enabled scheme and the notation D denotes the traditional case.

**Figure 6 entropy-25-00307-f006:**
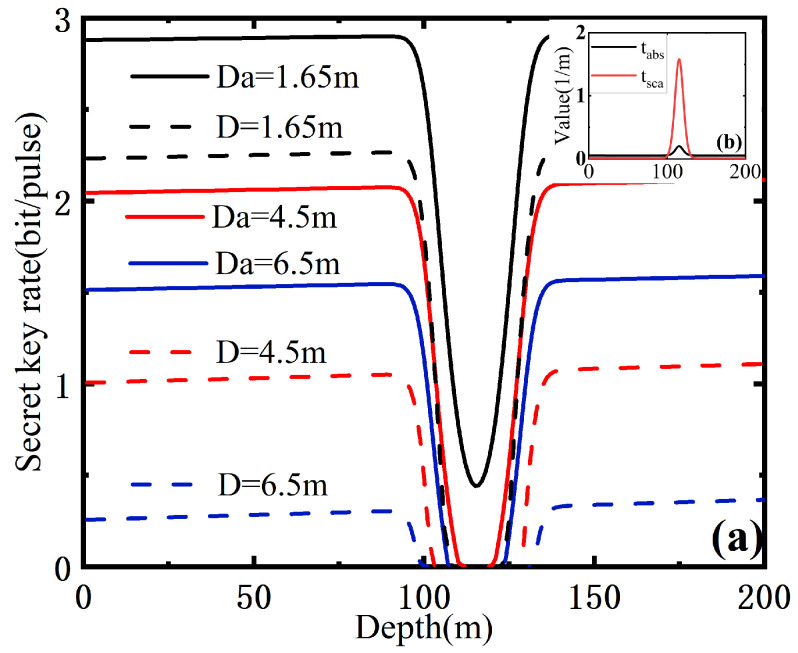
The secret key rate of the PO-enabled CVQKD system as a function of the depth. The notation Da denotes the case of the PO-enabled scheme and the notation D denotes the traditional case.

## Data Availability

All data generated or analyzed during this study are included in this published article.

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
