# Peer review of "Passive Continuous Variable Quantum Key Distribution through the Oceanic Turbulence"

_entropy, 2023, doi:10.3390/e25020307_

Round 1
Reviewer 1 Report
The authors explore the passive continuous variable quantum key distribution through oceanic turbulence, which shows a certain potential in future chip integration and portable quantum key distribution. However, there are some modifications needed before its publication:
1. In line 72 on Page 2, the authors claim that Bob’s actions on mode B_1 can be performed as same before. However, the corresponding Figure 1 shows that there is a PO module in Bob’s side, which is a significant module for performance improvement of the system mentioned in the manuscript.? Is it the same as that in the GMCS protocol or that in other systems that use PO operations? I think more explanations and references are needed here.
2. In the caption of Figure 3, the authors claim that the blue line denotes the result after using the AO unit. What is the AO unit here? Please check it.
3. As it shown in Figure 3, the blue line still coincides with the gray line for a period of time, which means that the PO unit will not start working until a very short period of time. What is the reason for this phenomenon?
4. Overall the flow of the writing is good, but small issues with e.g. verb tense, grammar, and case need more attention. Please check and modify the manuscript.
5. The following references should be included in [11-13]:
-Z. Chen, Y. Zhang, G. Wang, Z. Li, and H. Guo, Phys. Rev. A 98, 012314 (2018).
-Z. Chen, Y. Zhang, X. Wang, S. Yu, and H. Guo, Entropy 21, 652 (2019).
Reviewer 2 Report
The work deals with a current and interesting topic. In this sense, it is notable that the authors invested significantly in the development and analysis of results. In terms of results, the authors carried out a very interesting analysis in Figure 4. Therefore, it was possible to perceive that the authors carried out an analysis of precursor works that were far behind in the related literature, which even presents only 19 works. as a suggestion to compose the literature base.
Scarani, V., Bechmann-Pasquinucci, H., Cerf, N. J., Dušek, M., Lütkenhaus, N., & Peev, M. (2009). The security of practical quantum key distribution. Reviews of modern physics, 81(3), 1301.
Lira, E. R., de Macêdo, H. B., Lima, D. A., Alt, L., & Oliveira, G. (2021). A reversible system based on hybrid toggle radius-4 cellular automata and its application as a block cipher. arXiv preprint arXiv:2106.04777.
Fedorov, A. K., Kiktenko, E. O., & Lvovsky, A. I. (2018). Quantum computers put blockchain security at risk.
Chakraborty, K., Rozpedek, F., Dahlberg, A., & Wehner, S. (2019). Distributed routing in a quantum internet. arXiv preprint arXiv:1907.11630.
Reviewer 3 Report
This paper analyzes a CWQKD scheme with novel features that does not appear to have been considered before. It proposes using a passive state preparation scheme based on thermal sources for the key generation rather than an active state scheme that would be far more technically demanding. In many QKD schemes that involve transmission through the atmosphere, atmospheric turbulence presents a major obstacle that is not easy to deal with. In the present scheme there is seawater turbulence to be compensated for, and the authors propose using a PO unit to address this task. Through numerical solutions the authors examine the merits of their scheme in terms of the secret key rate, transmission distance and depth. Many results of the simulation are summarized in Fig.5, and some advantages of the PO scheme over the traditional one are pointed out. The present scheme seems to fill a void in the THz band, and illustrates the sort of advantages that can be made use of in the seawater environment. I think this scheme deserves a wider discussion and so I would recommend publication of the paper.
